# Mechanistic manifold in a hemoprotein-catalyzed cyclopropanation reaction with diazoketone

Donggeon Nam[1,6], John-Paul Bacik[2,6], Rahul L. Khade[3,6], Maria Camila Aguilera[1], Yang Wei [3], Juan D. Villada[1,5], Michael L. Neidig [4] ✉, Yong Zhang [3] ✉, Nozomi Ando [2] ✉ & Rudi Fasan [1,5] ✉

Hemoproteins have recently emerged as promising biocatalysts for new-to-nature carbene transfer reactions. However, mechanistic understanding of the interplay between productive and unproductive pathways in these processes is limited. Using spectroscopic, structural, and computational methods, we investigate the mechanism of a myoglobin-catalyzed cyclopropanation reaction with diazoketones. These studies shed light on the nature and kinetics of key catalytic steps in this reaction, including the formation of an early heme-bound diazo complex intermediate, the rate-determining nature of carbene formation, and the cyclopropanation mechanism. Our analyses further reveal the existence of a complex mechanistic manifold for this reaction that includes a competing pathway resulting in the formation of an N-bound carbene adduct of the heme cofactor, which was isolated and characterized by X-ray crystallography, UV-Vis, and Mössbauer spectroscopy. This species can regenerate the active biocatalyst, constituting a non-productive, yet non-destructive detour from the main catalytic cycle. These findings offer a valuable framework for both mechanistic analysis and design of hemoprotein-catalyzed carbene transfer reactions.

Transition metal-catalyzed carbene transfer reactions constitute a powerful strategy for forging new carbon-carbon and carbon-heteroatom bonds[1–5]. While these transformations have been traditionally addressed using organometallic catalysts[1–5], recent efforts have made possible the realization of these transformations using engineered enzymes[6–8]. In particular, heme-containing proteins and enzymes such as cytochrome P450s, myoglobin, and cytochrome *c* proteins, have provided promising metalloprotein scaffolds for catalyzing a growing number of carbene transfer reactions, including cyclopropanation[9–14], Y-H carbene insertion (Y = N, S, B, Si)[15–19], C–H insertion[20–23], and others, with high activity and stereocontrol.

As protein engineering efforts have expanded the scope of hemoprotein-catalyzed carbene transfer reactions, recent efforts have also begun to shed light on the mechanism and catalytic intermediates of some of these transformations. In this context, the mechanism of hemoprotein-catalyzed cyclopropanation with diazoesters has been investigated computationally and experimentally, supporting the intermediacy of a heme-bound carbene as a key catalytic intermediate for the carbene transfer step[24,25]. Notably, using an engineered cytochrome *c* variant and a diazopropanoate ester reagent, Lewis et al. were able to trap the heme-bound carbene intermediate implicated in the Si-H (and B-H) carbene insertion reaction catalyzed by this

[1]Department of Chemistry, University of Rochester, Rochester, NY 14627, USA. [2]Department of Chemistry and Chemical Biology, Cornell University, Ithaca, NY 14853, USA. [3]Department of Chemistry and Chemical Biology, Stevens Institute of Technology, Hoboken, NJ 07030, USA. [4]Inorganic Chemistry Laboratory, Department of Chemistry, University of Oxford, South Parks Road, Oxford OX1 3QR, UK. [5]Present address: Department of Chemistry and Biochemistry, University of Texas at Dallas, Richardson, TX 75080, USA. [6]These authors contributed equally: Donggeon Nam, John-Paul Bacik, Rahul L. Khade. ✉e-mail: michael.neidig@chem.ox.ac.uk; yong.zhang@stevens.edu; nozomi.ando@cornell.edu; rudi.fasan@utdallas.edu

enzyme[26]. This species was determined to have a singlet ground electronic state in line with previous computational analyses using a myoglobin-based cyclopropanation catalyst[24,25]. Using a myoglobin-based cyclopropanase[27] bearing a non-proteinogenic N-methyl-histidine as an axial ligand, Hilvert and coworkers isolated a long-lived heme-carbene adduct with a bridging Fe(III)–C–N(pyrrole) configuration[28]. While catalytically inactive toward cyclopropanation, this species was proposed to equilibrate with the reactive heme-carbene complex, thus representing an inert, yet reversible off-cycle intermediate[28]. A few other biocatalytic carbene transfer reactions have also been the object of mechanistic investigations[15,17,29–31].

Despite this progress, our understanding of the catalytic mechanism and intermediates implicated in this emerging class of enzymatic reactions remains limited. Here, we applied a combination of complementary spectroscopic, crystallographic, and computational tools to investigate and elucidate the mechanism and catalytic steps of a recently reported myoglobin-catalyzed cyclopropanation reaction with diazoketones[32]. From these studies, important insights into the kinetic, spectroscopic, and structural features of key catalytic steps and intermediates formed in this reaction were obtained. Importantly, our studies demonstrate the occurrence of a complex interplay of productive and competing pathways during catalysis, thus revealing a significantly more intricate mechanism than assumed based on previously available studies.

## Results and discussion
### Isolation and structural characterization of myoglobin-carbene adduct from reaction with diazoketone
We recently reported the development of a biocatalytic strategy for the cyclopropanation of styrene (**1**) and vinylarenes with benzyl diazoketone (**2**, BDK)[32]. An optimal biocatalyst for this reaction was established to be myoglobin (Mb) variant Mb(H64G,V68A), which delivers the (1S,2S) cyclopropanation product **3** with excellent

stereoselectivity (>99% de and ee) and up to ~1000 catalytic turnovers (TON) (Fig. 1a). As determined previously, optimal reaction conditions for this cyclopropanation reaction (in terms of TON) involve the use of ferrous Mb as the catalyst, produced from the ferric form of the protein via in situ reduction with sodium dithionite, and a 4:1 ratio of styrene: BDK[32]. During the investigation of this reaction, we noted that the protein undergoes a peculiar change in color from the characteristic red color to green (referred to here as 'Mb-cIII' species) upon exposure to the BDK reagent in the absence of the styrene substrate (Fig. 1b). The green color appears after 1–2 min after addition of the diazo reagent and persists for hours before returning to red. In the presence of styrene, no color change is observed. As determined by UV-Vis absorption spectroscopy, this color change corresponds to a change in a Soret band peak from 430 nm to a lower intensity Soret band at 427 nm (Fig. 1b). The Q-band region is also affected, showing a reduction in absorbance at 530/560 nm and an increase in intensity at 590 nm.

Intrigued by these observations and to gain insights into the function of myoglobin-based carbene transferases, we pursued structural elucidation of the green Mb-cIII species via freeze-trapping X-ray crystallography. When crystals of Mb(H64G,V68A) are soaked with the BDK reagent, the color turns from red to green over the course of a few minutes to about an hour, depending on the crystal size. Soaked crystals were trapped by rapid plunging in liquid nitrogen at various time points, and diffraction and UV-Vis absorption data were collected on the same crystals. After optimization of the soaking and incubation conditions, a structure of the green complex was determined to a resolution of 1.40 Å (Fig. 2a, Supplementary Table 21). The Mb(H64G,V68A) variant can also be isolated as an imidazole-bound complex, due to its active site mutations resulting in a high affinity for imidazole ($K_D = 30$ µM vs. 102 mM for wild-type Mb; Supplementary Fig. 2). Imidazole-bound Mb(H64G,V68A), hereafter referred to as Mb-imi, exhibits identical reactivity in the cyclopropanation reaction as the

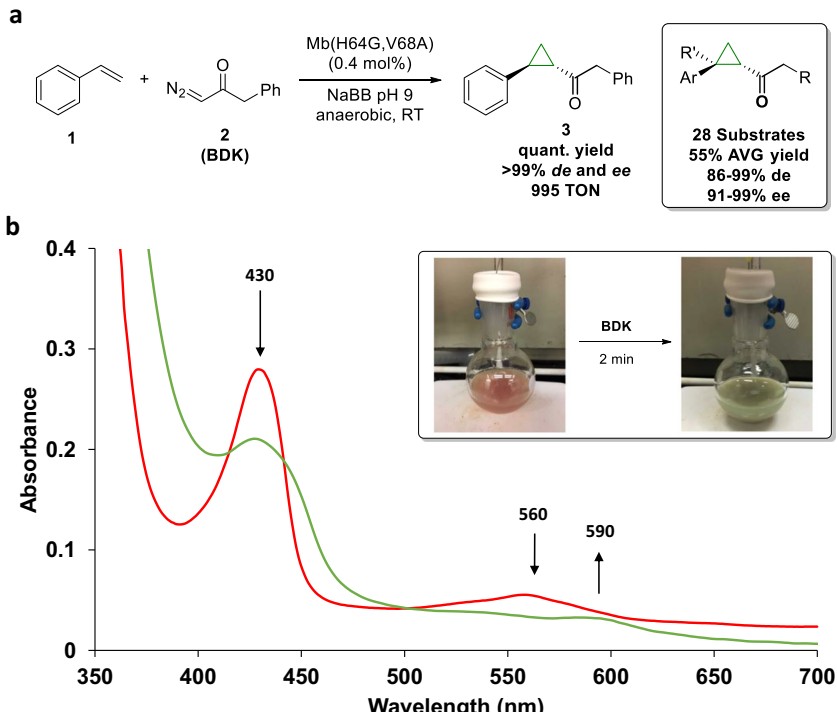

**Fig. 1 | Mb reaction with benzyl diazoketone. a** Mb(H64G,V68A)-catalyzed cyclopropanation of styrene with benzyl diazoketone (BDK) and substrate scope of the biocatalyst (*box*; R = aryl or alkyl; R' = H or Me). **b** Color change of a Mb(H64G,V68A) solution upon addition of BDK (*box*) and corresponding transition in the UV-Vis absorption spectrum (2 mM Mb(H64G,V68A)) in 50 mM sodium borate buffer (NaBB) (pH 9.0), anaerobic conditions, before (red curve) and after (green curve) addition of BDK.

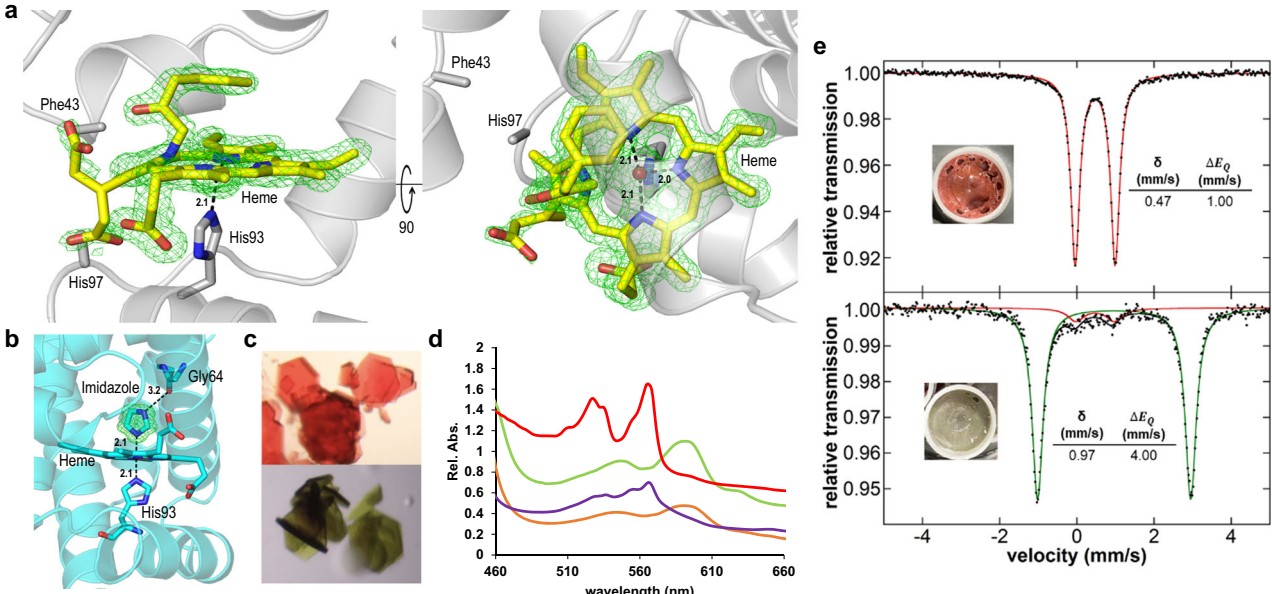

**Fig. 2 | Crystallographic and spectroscopic characterization of Mb(H64G, V68A). a** Mb-cIII and **b** Mb-imi structures are shown with proximal histidine and ligands in stick format. mF$_o$-DF$_c$ omit electron density maps (green mesh) are shown at 3σ, distances are in angstroms. Positions of disordered residues Phe43 and His97 are also shown. **c** Crystals of myoglobin in the presence of imidazole (top) or BDK (bottom). **d** Overlay of the Q-band regions of the UV-Vis absorption spectra of Mb-imi and Mb-cIII acquired in solution (purple and orange, respectively) and in crystallo (red and green, respectively). In crystallo spectra are normalized (Abs./2) to facilitate comparison. **e** 80 K $^{57}$Fe Mössbauer spectra of ferrous Mb-imi (3 mM protein, after incubation with 30 mM Na$_2$S$_2$O$_4$ for 30 s; red component) and Mb-cIII, trapped after further incubation with 100 mM BDK for 30 s (green component).

free enzyme (Supplementary Table 1, Supplementary Fig. 6) and it was found to be better suited for Mössbauer experiments (vide infra). We therefore also determined the crystal structure of ferric Mb(H64G,V68A) in complex with imidazole to 1.04 Å resolution (Fig. 2b). The color change of representative crystals of the two states can be observed in Fig. 2c.

To our surprise, structural characterization of Mb-cIII compound revealed that this species corresponds to a protein complex in which the BDK-derived carbene moiety is transferred to a ring nitrogen atom of the porphyrin cofactor (Fig. 2a). As a result of this modification, the heme cofactor is distorted with the pyrrole ring C lying out of plane at an angle of ~30° compared to the planar heme. One of the heme propionate groups, which is typically located on the distal side of the heme, shows dual occupancies, with an alternate conformation of the propionate group being translocated to the proximal side of the heme. Phe43 and residues adjacent to it appear disordered, presumably due to a steric clash with the phenyl moiety of the inserted compound. Furthermore, His97 and adjacent residues at the proximal side of the heme also appear disordered due to the translocation of the propionate group. In our structure, the α-carbon insertion adopts a tetrahedral geometry, indicating protonation of the carbon atom following insertion. Furthermore, the observed interatomic distance between the α-carbon atom and Fe center in Mb-cIII is 2.7 Å, which is considerably longer than that observed for the C–Fe bond length (1.9 Å) in a Fe-N bridged heme-carbene complex[28]. While similar N-alkylated heme complexes with free metalloporphyrins have been reported[33,34] and N-alkylated porphyrin adducts have been observed from reactions of microsomal P450s with halogenated and olefinic compounds[35–37], to our knowledge, the isolation and structural elucidation of such complex in the context of a protein is unprecedented. Importantly, the correspondence between the green complex formed in solution and the crystallized protein complex was confirmed by matching of the Q-band regions as determined via in crystallo UV-vis spectroscopy (Fig. 2d).

The crystal structure of Mb(H64G,V68A) exhibits an overall similar fold with that of wild-type Mb as well as a related variant, Mb(H64V,V68A), investigated previously[25], with an RMSD of 0.14 Å

calculated using PyMOL (all-atoms with outlier removal). As a key difference, Mb(H64G,V68A) shows a significantly larger opening connecting the solvent to the heme pocket due to the substitution of the distal His residue (His64) with a glycine. The H64G mutation expands the volume of the heme distal pocket to 339 Å$^3$, compared to 243 Å$^3$ in Mb(H64V,V68A) and 125 Å$^3$ for the wild-type myoglobin structure[25]. Of note, the reduction of steric hindrance at this position was critical for enhancing the enzyme reactivity in this reaction (His « Val < Ala < Gly)[32], and this trend can now be rationalized based on the effect of the H64G mutation in providing a more facile access of the bulky diazo reagent to the iron center. The enhanced binding of imidazole may also be rationalized through a hydrogen bond between a ring nitrogen of imidazole and the main chain oxygen of the mutated glycine residue (Fig. 2b). Previous structures of wild-type ferric sperm whale Mb also showed that the distal histidine residue must swing out of its position in the pocket to accommodate imidazole binding[38].

## Characterization by Mössbauer spectroscopy

The Mb-cIII complex, along with the Mb-imi complex, were further characterized by $^{57}$Fe 80 K Mössbauer spectroscopy (Fig. 2e). For these experiments, $^{57}$Fe-labeled Mb(H64G,V68A) was reduced with sodium dithionite, followed by addition of BDK, and the resulting speciation distribution was trapped by freeze-quenching (Fig. 2e). The Mössbauer spectrum of Mb-imi supports the presence of a low-spin iron (II) species with an isomer shift (δ) of 0.47 mm/s and a quadrupole splitting (|ΔE$_Q$|) of 1.00 mm/s. DFT calculations (see Supplementary Information for details) indicated that these experimental values best match with a singlet (S = 0) form of the ferrous imidazole complex (Supplementary Table 2). By contrast, the Mössbauer spectrum of Mb-cIII revealed that this species corresponds to a high-spin iron (II) with an isomer shift (δ) of 0.97 mm/s and a quadrupole splitting (|ΔE$_Q$|) of 4.00 mm/s (Fig. 2e). DFT calculations were carried out to predict the Mössbauer parameters for the Mb-cIII species observed by crystallography, as well as other possible complexes such as heme-bound carbene[24,26] and Fe–N bridging carbene[28] forms analogous to those previously isolated from reactions between other hemoproteins with diazoesters. A best match between

the experimental and simulated Mössbauer parameters was found for the quintet (S = 2) form of the Mb-cIII complex observed by crystallography (Supplementary Table 2). In addition, compared to other spin states, the DFT-optimized quintet state of the Mb-cIII model (called $P_{N-alkylation}$) corresponds to the lowest energy spin state and it presents the best match with the N-alkylated heme complex in the crystal structure, as judged based on its smallest mean percent deviation (MPD) of key bond lengths and angles around iron (MPD of 2.3% vs. 2.8% and 5.5% for S = 1 and S = 0, respectively; Supplementary Table 16). In particular, for this lowest energy structure, the MPD for bond distances (which have a greater effect on energy than bond angles) is only 1.4% (with a mean absolute deviation of only 0.03 Å). Altogether, these analyses provide insight into the electronic structure of the newly identified N-alkylated heme protein complex.

## Mass spectrometry analysis

Heme extraction experiments were performed to detect the N-alkylated heme adduct by mass spectrometry (MS). MS analysis of acetone/HCl extracts of Mb(H64G,V68A) after brief exposure with BDK showed isolation of a species with a m/z ratio and isotopic distribution consistent with the expected N-alkylated protoporphyrin IX adduct after acid-induced demetallation (Supplementary Fig. 10), as previously observed for N-alkylated complexes derived from cytochromes[35–37]. In addition, a signal consistent with the m/z of the bridged carbene-heme complex, which is expected to mediate the formation of Mb-cIII from the heme-carbene complex (vide infra), was also observed, thus providing further experimental evidence for the formation of both of these species in the enzymatic reaction (Supplementary Fig. 10).

## Mechanistic scenarios

While cyclopropanation reactions by Mb and other hemoproteins are thought to be mediated by a heme-carbene complex[24], isolation and characterization of the Mb-cIII complex raised intriguing questions about its involvement in the catalytic cycle of this reaction. Accordingly, different mechanistic scenarios involving this species were formulated to guide further experiments (Fig. 3). The first one views Mb-cIII as a 'dead-end' inactivation byproduct (Fig. 3a), as suggested by previous reports on related complexes obtained from iron-porphyrin catalysts[33]. Two alternative scenarios implicate the Mb-cIII complex as an 'off-cycle' intermediate derived from the heme-carbene intermediate (or a precursor thereof). In one case (Fig. 3b), the Mb-cIII complex is a cyclopropanation incompetent species that is able to regenerate the heme-carbene complex for cyclopropanation, i.e., playing a role akin to a Fe–N bridging heme-carbene complex recently reported by Hilvert and coworker[28]. In another case, the Mb-cIII complex re-enters the catalytic cycle through regeneration of the Mb enzyme (Fig. 3c). Albeit less likely, we also considered a scenario in which the Mb-cIII complex is catalytically competent toward cyclopropanation and contributes, in place of or along with the heme-carbene intermediate, to the cyclopropanation reaction (Fig. 3d). Based on these working hypotheses, experiments were conducted to probe these different mechanistic scenarios.

## Kinetic analysis of the cyclopropanation reaction

To gain insights into the kinetics of catalytic steps and reaction intermediates, the reaction of Mb(H64G,V68A) with BDK was further analyzed using stopped-flow UV-Vis spectroscopy (Fig. 4). These experiments established that reduction of the hemoprotein from the ferric to the catalytically active ferrous form in the presence of sodium dithionite is very fast, with this step being characterized by a second order constant of $1.4 \pm 0.1 \times 10^8$ M$^{-1}$ s$^{-1}$, as derived from plots of $k_{obs}$ vs. [Na$_2$S$_2$O$_4$] under pseudo-first-order conditions (Supplementary Fig. 3). This value is comparable to previously reported rate constants for sodium dithionite reduction of wild-type Mb ($6 \times 10^7$ M$^{-1}$ s$^{-1}$)[39]. Starting from ferrous Mb(H64G,V68A), the addition of the BDK reagent results

in two major changes in the UV-vis absorption spectrum of the protein, the first one occurring within a short timescale (<2 s; Fig. 4a), followed by a second, slower transition (over 2–4 min; Fig. 4b) resulting in the accumulation of a long-lived species with spectroscopic features identical to that of the Mb-cIII complex.

The fast transition is characterized by a blue shift of the Soret band from 435 nm, corresponding to the ferrous protein, to a catalytic intermediate with a Soret peak at 425 nm, referred to as Mb-cI. This step exhibits a first-order dependence on BDK concentration with a second-order rate constant of $4.0 \pm 0.4 \times 10^9$ M$^{-1}$ s$^{-1}$, as determined via titration experiments (Fig. 4a). In contrast, the following transition, which is characterized by an isochromic reduction in intensity of the Soret band and concomitant decrease and increase of Q bands at 530 nm 590 nm, respectively, shows a zero-order dependence on BDK concentration (Fig. 4b). For this step, a first-order rate constant of $7900 \pm 74$ s$^{-1}$ was derived from analysis of the exponential decay of intermediate Mb-cI over time. These experiments thus indicated that the N-alkylated protein complex Mb-cIII is formed from the earlier intermediate Mb-cI via a unimolecular reaction. Further monitoring of the reaction showed that while the Mb-cIII is long-lived ($t_{1/2}$ ~ 20 min), this intermediate eventually disappears over 2–3 h to give rise to a species with spectral features identical to that of Mb (Supplementary Fig. 5A). Thus, these results provided initial evidence that Mb-cIII is not a dead-end intermediate (Fig. 3a), but rather an off-cycle or catalytic intermediate in line with mechanisms in Fig. 3b, c, or d.

Stopped-flow kinetic analyses were then conducted in the presence of varying concentration of styrene (3–20 mM) with BDK at a fixed concentration (1 mM), which revealed a large impact of the olefin concentration on the evolution of the reaction intermediates over the 300-s time window (Fig. 4c, d and Supplementary Fig. 4). Specifically, at low styrene concentration, the reaction is dominated by the accumulation of Mb-cIII, as observed in the absence of the olefin (Fig. 4c). As the styrene concentration increases, Mb-cIII formation is progressively outcompeted by the accumulation of a species with spectral features corresponding to the ferrous Mb to a point ([styrene] > 10 mM), where formation of Mb-cIII is not generated anymore (Fig. 4d and Supplementary Fig. 4). Similar results were obtained from a double mixing experiment in which styrene was added after blue shift of the Soret band induced by addition of the diazo compound (Fig. 4a). It is worth noting that the experiments in the presence of excess styrene over the diazo compound (i.e., [styrene] > 10 mM) closely mimic the optimized conditions applied for synthesis[32] and that, under these conditions, the cyclopropanation reaction proceeds with an initial turnover frequency (TOF) of 200 min$^{-1}$ and it reaches completion within 2 min, as determined from time-course experiments (Supplementary Fig. 6). Since Mb-cIII is formed at a longer timescale (2–4 min) than the cyclopropanation reaction (<1–2 min), these results disfavor the possibility of Mb-cIII being involved in cyclopropanation as catalytic species (Fig. 3). Rather, the results described in Fig. 4c, d and Supplementary Fig. 6 are consistent with mechanisms where Mb-cIII is an off-cycle intermediate whose formation competes for cyclopropanation (i.e., Fig. 3b or c; Supplementary Fig. 4A). Another important conclusion from these studies is that the species observed at high olefin concentration represents the resting state of the enzyme under conditions and timescales (i.e., 100–300 s) allowing for multiple catalytic turnovers.

## Role of Mb-cIII as off-cycle intermediate

Further studies were then carried out to further dissect the role of Mb-cIII in the reaction and discern between mechanistic scenarios B and C of Fig. 3. To this end, time-course Mössbauer experiments were conducted to monitor the formation and evolution of Mb-cIII over time. Consistent with the results from the stopped-flow experiments, these analyses showed, upon addition of BDK, the hemoprotein is converted into the Mb-cIII complex over the course of 3 min and then the latter slowly converts back to Mb over 4 h in the absence of styrene

**Fig. 3 | Different mechanistic scenarios involving Mb-cIII in Mb-catalyzed cyclopropanation with BDK.** IPC iron-porphyrin carbene complex. **a** Mechanism featuring Mb-cIII as dead-end byproduct. **b** Mechanism with Mb-cIII as off-cycle intermediate capable of re-generating IPC. **c** Mechanism featuring Mb-cIII formation from IPC as an alternative pathway competing with olefin cyclopropanation by IPC. **d** Mechanism with Mb-cIII as catalytically competent intermediate capable of mediating olefin cyclopropanation.

(Supplementary Fig. 7). In addition to further supporting the conclusion that the Mb-cIII complex is not a catalytic end-point, these experiments showed that this species can regenerate catalytically active hemoprotein in the absence of the olefin substrate and, thus, they support its role as an off-cycle intermediate that is not in equilibrium with the heme-carbene complex implicated in cyclopropanation (i.e., Fig. 3b). This conclusion is further supported by additional experiments showing a reduction in cyclopropanation product yield upon pre-incubation of Mb(H64G,V68A) with BDK prior to addition of styrene (Supplementary Fig. 8).

## Quantum chemical calculations and mechanism of cyclopropanation

Density Functional Theory (DFT) studies were carried out to provide a framework for understanding the different reaction pathways implicated in this transformation. As shown in Fig. 5, these calculations utilized a histidine-ligated (modeled as 5-methylimidazole) Fe-porphyrin ($R_{heme}$) to mimic the myoglobin biocatalyst, BDK as the

reactant, and methods established previously for accurate prediction of heme-catalyzed carbene reactions[24,25,31,40–42] (see Supplementary Information for details). Because $R_{heme}$ and the heme-bound carbene (IPC) have quintet and singlet ground states as reported previously[41], our analyses focused on these two spin states. Our calculations show that the most favorable pathway for the cyclopropanation reaction proceeds via an initial carbon-bound diazo-heme complex ($Int_{C1}$), followed by an IPC intermediate ($Int_{IPC}$), which reacts with the olefin substrate to give the cyclopropane product (Fig. 5 and Supplementary Fig. S12). Unlike the lower-energy quintet state, which showed no binding to the iron, the singlet state $Int_{C1}$ ($^1Int_{C1}$) shows a weak coordination to the iron center (Fe-C distance: 2.42 Å) and remains readily accessible ($\Delta G = +8.9$ kcal/mol vs. $R_{heme}$). In addition, $^1Int_{C1}$ was found to be connected to the singlet state of $^1Int_{IPC}$ and corresponding transition state, $^1TS_{Carbene}$——which are more stable than the respective quintet states ($\Delta G = 7.2$ and 13.2 kcal/mol, respectively)——, via an intrinsic reaction coordinate (IRC) (see Supplementary Fig. 13). These data suggest that a spin crossover at the level of $Int_{C1}$ is required for the

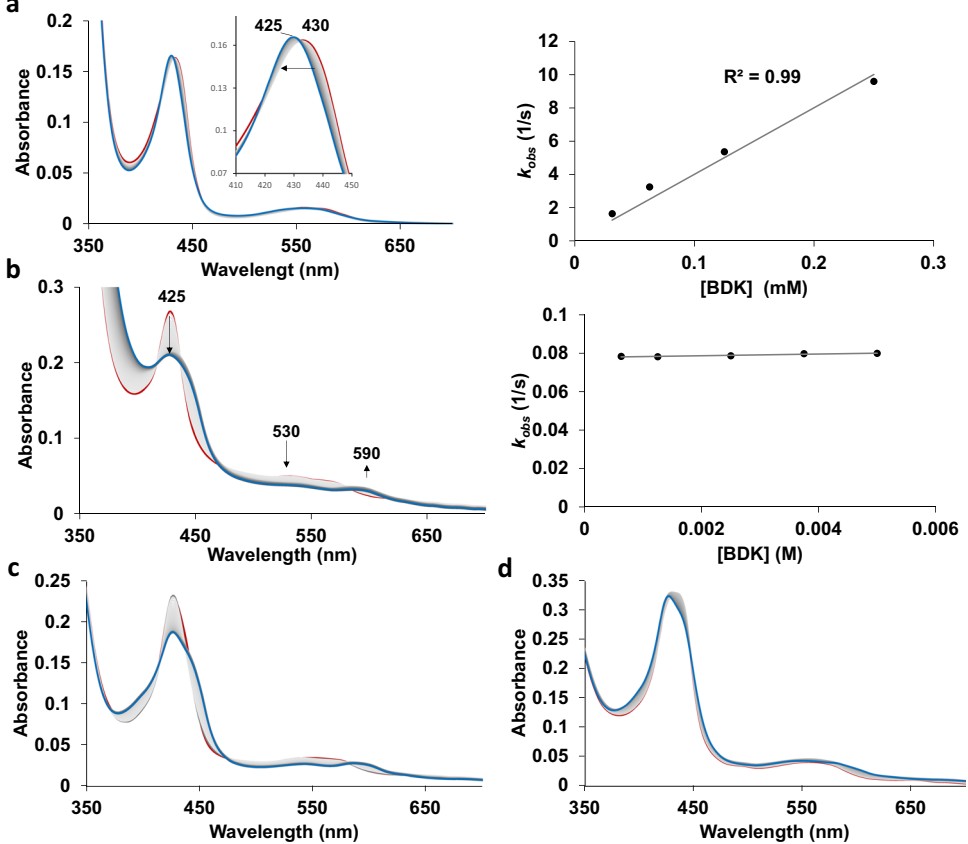

**Fig. 4 | Stopped-flow UV-Vis analysis of Mb-catalyzed cyclopropanation of diazoketone. a** UV-vis absorption spectra of the reaction of ferrous Mb(H64G,V68A) with BDK (1 mM) observed every 0.05 s for 1 s (red line: 0.05 s; blue line: 1 s) and plot of apparent rate constant ($k_{obs}$) against BDK concentration; **b** UV-Vis spectra of the same reaction as in (**a**) observed from 1 s to four min every 1 s after addition of BDK (red line: 1 s; blue line: 240 s) and plot of apparent rate constant against BDK concentration; **c**, **d** UV-Vis spectra of the reaction of ferrous Mb(H64G,V68A) with BDK (1 mM) in the presence of styrene at either **c** low (3 mM) or **d** high concentration (20 mM) every 2 s over 4 min. See Supplementary Figs. 3 and 5 for additional data.

next step (carbene formation). In $^1TS_{Carbene}$, the C-$N_b$ bond (C is carbene's carbon and $N_b$ is the nitrogen bonded to C) is 0.54 Å longer than in BDK, showing partial release of $N_2$, while the Fe-C distance is further decreased by 0.47 Å compared to $^1Int_{C1}$, showing bond formation (Supplementary Fig. 12). In $^1Int_{IPC}$, the Fe-C bond length is further reduced by 0.20 Å and a significant charge transfer of $-0.309\ e$ occurs from carbene to iron, leading to a positively charged carbene carbon center posed for nucleophilic attack by the olefin (Supplementary Fig. 12).

For the cyclopropanation step, a concerted pathway was considered on the basis of experiments supporting a non-radical cyclopropanation mechanism, as determined by lack of product inhibition in the presence of the spin-trapping reagent DMPO and absence of Z → E isomerization with deutero-styrene as the substrate (Supplementary Tables S3 and S4). In this step, the $^1Int_{IPC}$ intermediate reacts with styrene to form the cyclopropane product via transition state $TS_{CP}$ in the singlet state, which is preferred over its quintet state by 8.6 kcal/mol. The structural features of $TS_{CP}$ indicate an early transition state and a concerted but nonsynchronous carbene addition to the olefin (Fig. 5 and Supplementary Fig. S12). Importantly, these analyses established that the transition state for the cyclopropanation step lies at a lower energy than that of the carbene formation step ($\Delta G^{\ddagger} = 11.8$ vs. 24.4 kcal/mol), indicating that the latter constitutes the rate-determining step in this pathway.

## Diazo-heme complex formation

Given the relevance of the findings above for interpretation of the early intermediate Mb-cI observed spectroscopically (Fig. 4a), other possible diazo-heme coordination modes (and cyclopropanation pathways) were analyzed, involving an N-bound complex ($Int_{N1}$), where BDK is bound to the heme iron via the diazo group, and an O-bound complex ($Int_{O1}$), in which BDK coordinates to the heme iron via the carbonyl group. None of these heme-diazo-heme binding modes have been considered in prior mechanistic studies of biocatalytic carbene transfer reactions by us or other groups. While key results are summarized here, a detailed description of these analyses is provided in the Supplementary Information. Interestingly, BDK binding to the heme via the N-bound and O-bound modes were found to be both viable, with calculated ΔG values of 8.6 and 4.9 kcal/mol for $Int_{N1}$ and $Int_{O1}$, respectively (orange and green pathways, Fig. 5; Supplementary Figs. 15–17). $Int_{N1}$ is able to bind the iron center only in the singlet state (Supplementary Fig. 15), whereas $Int_{O1}$ can maintain Fe-O coordination in both singlet and quintet states (Supplementary Figs. S16 and S17). Compared to $Int_{C1}$, $Int_{N1}$ is nearly isoenergetic, while $Int_{O1}$ is lower in energy by ~4.0 kcal/mol, suggesting that all these binding modes are similarly viable. This notwithstanding, reaction pathways leading from $Int_{N1}$ or $Int_{O1}$ to the heme-carbene intermediate $Int_{IPC}$ were found to be associated with high energy barriers ($\Delta G^{\ddagger} > 40$ kcal/mol; Supplementary Figs. 15 and 17), which are comparable to that computed for the uncatalyzed reaction (Supplementary Fig. 14) and >15 kcal/mol higher than for $TS_{carbene}$ from $Int_{C1}$ (Fig. 5). Thus, it can be derived that while all three diazo-Fe binding modes (C-, N-, O-bound) are accessible, only $Int_{C1}$ is productive in leading to the IPC intermediate implicated in cyclopropanation. Given these results and the rate-determining nature of the $Int_{C1} \rightarrow Int_{IPC}$ being as discussed earlier, we assign the early intermediate Mb-cI to the Mb protein in the BDK-bound form. This

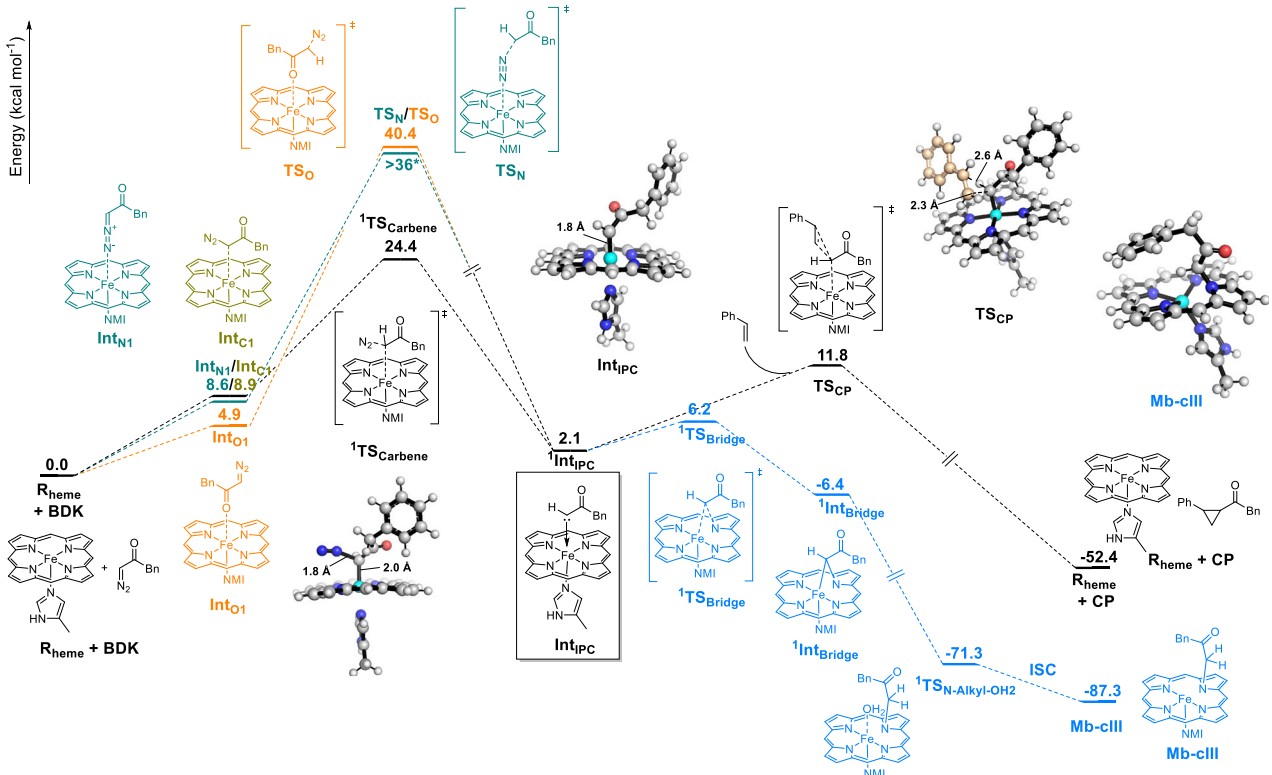

**Fig. 5 | Gibbs free energy diagram for the imidazole-bound heme with BDK and styrene.** The main cyclopropanation pathway is shown in black, with alternative/competing pathways colored in orange (O-bound heme-diazo complex formation), green (N-bound heme-diazo complex formation) and blue (Mb-cIII formation from IPC). ΔG values are calculated at the wB97XD/6−311G(d)-LANL2DZ level (see the Supplementary Information for details). NMI 5-methyl-imidazole. ISC intersystem crossing. * ΔG of TS_N could not obtained due to its highly unstable nature. Calculated ΔG of intermediate after TS_N is +36.4 kcal/mol indicating that ΔG(TS_N) must be >36 kcal/mol.

assignment is also consistent with this intermediate representing the resting state under catalytic conditions in the presence of styrene (Fig. 4e).

## Reaction pathway leading to Mb-cIII

Having elucidated the steps and intermediates implicated in the productive cyclopropanation process, a plausible pathway for the formation of Mb-cIII from heme-carbene complex (Int_IPC) was also investigated (Supplementary Fig. 20). A kinetically and thermodynamically feasible path was identified through the intermediacy of a bridging Fe−N carbene-heme complex (modeled as Int_Bridging), which can be formed from Int_IPC via a low energy barrier (ΔG‡ (TS_IPC-Bridge) = +4.1 kcal/mol). The formation of this species is corroborated by MS analyses of heme extraction samples (Supplementary Fig. 10). After reacting with a proton source such as $H_3O^+$, the Int_Bridge species can undergo protonation to give the Mb-cIII complex (modeled as P_N-alkylation; Supplementary Fig. 20). As for the Mb-cIII complex, Int_Bridging was found unable to promote cyclopropanation of styrene via both a concerted or radical mechanism (see Supplementary Information for further discussion), which is in accord to previous studies on similar complexes[28]. The path leading from Int_IPC to Mb-cIII is thermodynamically very favorable (ΔG = −89.4 kcal/mol), providing a strong thermodynamic driving force for the conversion from Int_Bridging to Mb-cIII. Interestingly, these calculations showed that, from Int_IPC, the energy barrier for the path leading to Mb-cIII is lower than that for cyclopropanation (ΔG = +4.1 vs. +11.8 kcal/mol), suggesting that (1) Mb-cIII formation is expected to be kinetically favorable in the absence of styrene and (2) competitive with cyclopropanation in the presence of it. These results are consistent with our experimental data demonstrating a dependence of Mb-cIII formation rate on styrene concentration (Fig. 4c, d and Supplementary Fig. 4).

## Biocatalyst regeneration from N-alkylated complex and other side reactions

Next, we focused on better understanding how catalytically competent protein is regenerated from the Mb-cIII complex, as evidenced by our UV-vis and Mössbauer experiments (Supplementary Figs. S5 and S7). To this end, we investigated (minor) reaction products generated upon incubation of Mb with BDK alone through reactions performed at high protein concentration (3 mM) to facilitate the accumulation and detection of these species. Insightfully, these reactions revealed the formation of phenyl propanone (**4**) as the major byproduct, along with 1-hydroxy phenyl propanone (**5**) and 1-methoxy phenyl propanone (**6**) (Fig. 6a, b). Phenyl propanone (**4**) apparently derives from the reduction of the diazoketone reagent, whereas compounds **5** and **6** can be attributed to O−H carbene insertion products with water and methanol, respectively, the latter being present in the reaction as co-solvent (10% v/v). Upon switching the methanol co-solvent with other alcohols (or DMSO), the reduction byproduct **4** and water insertion product **5** were still formed, while the alcohol insertion byproduct varied to match the alcohol present in the reaction (or absent in the case of DMSO), confirming the origin of this species (Supplementary Fig. 9). Time-course experiments were performed to assign these side reactions to catalytic steps and intermediates of the cyclopropanation reaction. These experiments show that the formation rate of the O-H insertion byproduct **7** correlates with the rate of formation of the cyclopropanation product (Fig. 6c), consistent with the hypothesis that this reaction (and alcohol O-H insertion) is mediated by the same reaction intermediate (i.e., IPC). Compared to **7**, the formation rate of the reduction byproduct **4** is >3-fold slower (0.14 TON/min vs 0.43 TON/min) (Fig. 6d) and it correlates instead with the rate of formation of Mb-cIII, as determined by UV-Vis (Fig. 4c). Based on these results, it can be derived that Mb-cIII is able to regenerate active Mb, and thus

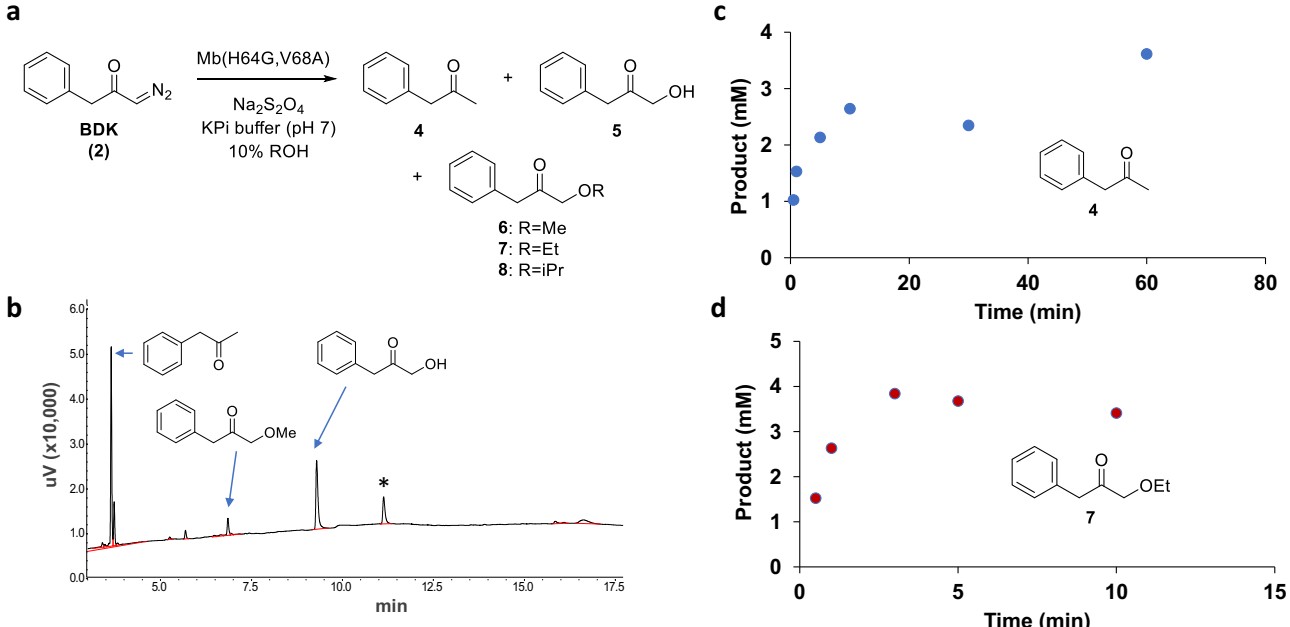

**Fig. 6 | Alternative reactions in Mb-catalyzed carbene transfer with BDK.**
**a** Alternative products generated from the reaction of Mb(H64G,V68A) with BDK in the absence of olefin and in the presence of different alcohols as cosolvents. **b** GC trace of reaction in the presence of 3 mM Mb(H64G,V68A), 30 mM $Na_2S_2O_4$, 100 mM BDK, in 50 mM potassium phosphate buffer (pH 7) with 10% MeOH, 16 h, room temperature, anaerobic conditions. See Supplementary Fig. 9 for additional data. * = unidentified product. **c**, **d** Time-course analysis of the formation of reduction byproduct 4 (**c**) and ethanol O-H carbene insertion product 7 (**d**) in the reaction with 3 mM Mb(H64G,V68A), 30 mM $Na_2S_2O_4$, 100 mM BDK, in 50 mM potassium phosphate buffer (pH 7), 10% (v/v) ethanol.

re-enter the catalytic cycle, via a reductive pathway that involves the release of **4** as byproduct. On the other hand, the O-H insertion reactions occur by the action of the earlier IPC intermediate, although these reactions are very inefficient (<2 TON; Fig. 6d).

**Further discussion**

By integrating the data and insights gathered from the spectroscopic, structural, and computational studies above, a mechanistic picture is proposed that encompasses both productive pathways and competing reactions in the myoglobin-catalyzed cyclopropanation of styrene with the diazoketone reagent. As shown in Fig. 7, the reduction of the hemoprotein to produce the catalytically active ferrous form is very fast ($k_2 = 1.4 \times 10^8$ $M^{-1}$ $s^{-1}$) and it is followed by another fast, bimolecular reaction ($k_2 = 4.0 \times 10^9$ $M^{-1}$ $s^{-1}$) resulting in the formation of the heme-bound diazoketone complex, which is characterized by a Soret band at 425 nm. Due to their small energy differences as determined by DFT, this species likely consists of a combination of the C-bound ($Int_{C1}$), N-bound ($Int_{N1}$) and O-bound forms ($Int_{O1}$) in equilibrium with each other, with the O-bound form being slightly more favorable energetically. As determined via DFT, only the C-bound complex is productive for carbene formation and cyclopropanation, and thus this constitutes the key intermediate leading to the subsequent catalytic steps (black path, Fig. 5). While the first step involving formation the diazo-heme complex is calculated to be endergonic ($\Delta G = +5$-9 kcal/mol), it remains readily accessible at room temperature and it is followed by an irreversible step involving diazo activation with $N_2$ extrusion (Fig. 5). In addition, it should be noted that the computed energies are derived from a stoichiometric reaction (i.e., 1:1 ratio) of the heme with the diazo reagent in order to compare all reaction pathways on the same footing. Experimentally, the large excess of BDK over the hemoprotein (mM vs. µM concentration) is expected to provide extra kinetic and thermodynamic driving force in the intermolecular reaction leading to the generation of the diazo-heme complex.

The subsequent diazo activation step to generate the heme-bound carbene intermediate has a calculated energy barrier ($\Delta G^{\ddagger}(^1TS_{carbene})$)

of 15.5 kcal/mol relative to the productive C-bound diazo-heme complex and it represents the rate-determining step in the overall cycle, as supported by our spectroscopic and computational analysis and consistent with the diazo-protein complex being the resting state under catalytic conditions (Fig. 4e). In stark contrast to the isolatable carbene-heme complex formed by cytochrome *c* with diazopropanoate ester[26], the heme-bound carbene complex mediating the cyclopropanation step in the present reaction could be neither isolated nor detected spectroscopically, indicating that it is highly reactive and short-lived. In the presence of styrene, the productive cyclopropanation reaction proceeds efficiently with an initial turnover frequency (TOF) of 200 $min^{-1}$. In the absence of styrene, or as the styrene concentration decreases due to its consumption, conversion of the heme-carbene intermediate to the N-alkylated complex Mb-cIII becomes prevailing. Interestingly, this step is kinetically more favorable than the cyclopropanation step and this along with our data in Fig. 4c, d, can explain why the heuristically optimized conditions for this reaction[32] involve an excess of olefin over the diazo compound. Indeed, these conditions are expected to favor the bimolecular cyclopropanation reaction between Mb-IPC and styrene over the unimolecular conversion of the Mb-IPC intermediate to Mb-cIII, also due to increased saturation of the enzyme. While the Mb-IPC could not be observed spectroscopically, it can be derived that the carbene formation step should approximate that measured for Mb-cIII formation from the diazo complex, i.e., $k \approx 8000$ $s^{-1}$. Since **Mb-IPC** does not accumulate in the absence of olefin, the subsequent step (Mb-IPC→Mb-cIII) must have a k > 8000 $s^{-1}$ (Fig. 7). In the presence of olefin, cyclopropanation outcompetes Mb-cIII formation (Supplementary Fig. 4); based on initial TOFs, it can be estimated that the cyclopropanation rate with 20 mM styrene is >400-fold faster than the rate of Mb-cIII formation from ferrous Mb (200 vs. 0.5 TON $min^{-1}$). Our mechanistic studies further indicate that the cyclopropanation step proceeds via a concerted, asynchronous carbene insertion into the olefinic double bond (Supplementary Tables S7 and S8, Supplementary Fig. 12), akin to that established for Mb-catalyzed cyclopropanation with diazoesters[24]. In addition, the present

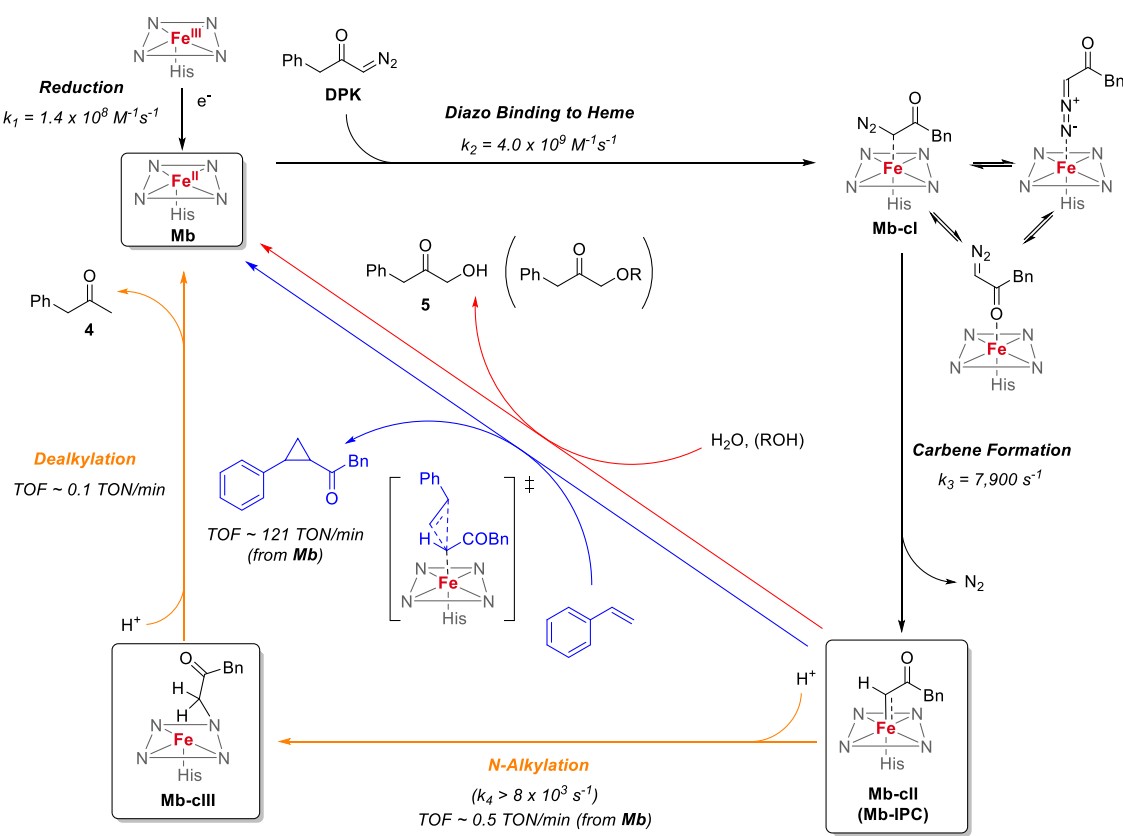

**Fig. 7 | Mechanistic manifold for Mb-catalyzed olefin cyclopropanation with benzyl diazoketone.** Global mechanistic picture describing productive and competing pathways in Mb-catalyzed styrene cyclopropanation with BDK as derived from the described mechanistic, spectroscopic, computational, and structural data. TOF turnover frequency.

studies establish that the carbene intermediate can also react with water and/or alcohol co-solvent to produce the corresponding O-H carbene insertion products. These side reactions, however, are inefficient (<1 TON) and not kinetically competitive over the cyclopropanation reaction, as they could be observed only under forced conditions and in the absence of styrene.

Our studies demonstrate that the Mb-cIII compound is not a catalytic dead-end, unlike what is suggested for related species isolated with metalloporphyrins[33], nor is it catalytically active, as it is in the case of a related N-alkylated complex formed upon reaction of Co-porphyrin with an acceptor-acceptor diazo compound recently reported by de Bruin and coworkers[34]. Rather, our studies collectively demonstrate that it is able to regenerate the biocatalyst via a path that, at least in part, involves the release of the reduction byproduct phenyl propanone. This overall step was estimated to occur with a rate of 0.1 TON min[−1], which is 5-fold slower than the rate of Mb-cIII formation from Mb. As a result, at low styrene concentration, the reaction is dominated by the accumulation of the green Mb-cIII complex, and biocatalyst regeneration becomes rate-limiting. Experimentally, we found that accumulation of the Mb-cIII species is observable not only in the absence of the olefin substrate (Fig. 1a) but also in the presence of unviable substrates. Accordingly, this phenomenon can be used as a proxy for successful diazo compound activation, but ineffective carbene transfer to the substrate by the hemoprotein, in the context of diazoketone reagents and possibly other diazo reagents. As a corollary of that, suppression of this competing pathway through rational design[43] could provide a means to enhance the desired carbene transfer reaction.

In conclusion, by a combination of spectroscopic, crystallographic, and computational analyses, this study provides new insights into the kinetics, rate-determining steps, and spectroscopic

and electronic features of key intermediates involved in a hemoprotein-catalyzed cyclopropanation reaction with diazoketones. These studies denote the mechanistic intricacies of this transformation, unveiling a complex interplay of productive and competitive reaction pathways while providing a rationale for understanding the improved catalytic activity of the system under heuristically optimized reaction conditions. Structural characterization of the novel Mb-cIII complex expands our knowledge of protein complexes formed upon the reaction of hemoproteins with diazo compounds[26,28] and carbene transfer products, revealing a dynamic structural and mechanistic manifold for such reactions. This information, along with the characterization of the spectral and kinetic features of key intermediates in this reaction, should provide a valuable framework for the mechanistic analysis of other carbene transfer reactions catalyzed by myoglobin and other hemoproteins. This mechanistic knowledge should also facilitate the design of novel and improved carbene transferases and thus facilitate efforts toward expanding the scope of biocatalysis in the context of abiological transformations of synthetic value.

## Methods
### General information
All the chemicals and reagents were purchased from commercial suppliers (Sigma-Aldrich, Alfa Aesar, ACS Scientific, Acros) and used without any further purification unless otherwise stated. All dry reactions were carried out under argon or nitrogen gas in flame-dried glassware with magnetic stirring using standard gas-tight syringes, cannula and septa. $^1$H, $^{13}$C, and $^2$H NMR spectra were measured on a Bruker DPX-400 instrument (operating at 400 MHz for $^1$H, 100 MHz for $^{13}$C, and 60 MHz for $^2$H) or a Bruker DPX-500 instrument (operating at 500 MHz for $^1$H and 125 MHz for $^{13}$C). Tetramethylsilane (TMS) served as the internal standard (0 ppm) for $^1$H NMR, CDCl$_3$ was used as

the internal standard (77.0 ppm) for $^{13}$C NMR and for $^2$H NMR (7.26 ppm). Silica gel chromatography purifications were carried out using AMD Silica Gel 60 230-400 mesh. Preparative thin-layer chromatography was performed on TLC plates (1 mm thickness, Sigma-Aldrich). All UV-Vis spectra were recorded on Shimadzu UV-2401PC at 20 °C with a 1 cm path length. Steady-state and pre-steady-state stopped-flow experiments were performed on an Applied Photophysics SX20 stopped-flow spectrophotometer (Applied Photophysics Ltd., Leatherhead, UK) equipped with a xenon arc lamp and a 1 cm path length.

### Protein expression and purification

The Mb variants were expressed in *E. coli* C41(DE3) cells[44]. Briefly, cells were grown in TB medium (ampicillin, 100 mg L$^{-1}$) at 37 °C (200 rpm) until OD$_{600}$ reached 1.0-1.2. Cells were then induced with 0.25 mM β-d-1-thiogalactopyranoside (IPTG) and 0.3 mM δ-aminolevulinic acid (ALA). After induction, cultures were shaken at 180 rpm and 27 °C and harvested after 18–20 h by centrifugation at 2800 × *g* at 4 °C. After cell lysis by sonication, the proteins were purified by Ni-affinity chromatography. The lysate was transferred to a Ni-NTA column equilibrated with Ni-NTA Lysis Buffer (50 mM potassium phosphate (KPi), 250 mM, NaCl, 10 mM imidazole, pH 8.0). The resin was washed with 50 mL of Ni-NTA Lysis Buffer and then 50 mL of Ni-NTA Wash Buffer (50 mM KPi, 250 mM, NaCl, 20 mM imidazole, pH 8.0). Proteins were eluted with Ni-NTA Elution Buffer (50 mM KPi, 250 mM, NaCl, 250 mM imidazole, pH 8.0). After elution, the proteins were buffer exchanged against 50 mM KPi buffer (pH 7.0) using 10 kDa Centricon filters. Protein concentration was determined by the CO-binding assay using $\varepsilon_{410} = 156$ mM$^{-1}$ cm$^{-1}$ as the extinction coefficient for UV-Vis spectroscopy.

For the Mössbauer experiments, $^{57}$Fe-labeled Mb variants were expressed in *E. coli* C41(DE3) cells grown in M9 minimal media (ampicillin, 100 mg L$^{-1}$). At an OD$_{600}$ of 0.6–0.8, cultures were centrifuged at 4000 rpm at 4 °C and the cell pellet was resuspended in $^{57}$Fe-enriched M9 minimal media containing 6 mg $^{57}$FeCl$_3$ and induced by the addition of 0.25 mM β-d-1-thiogalactopyranoside (IPTG) and 0.3 mM δ-aminolevulinic acid (ALA). After induction, cultures were grown at 27 °C for 18–20 h and the protein was purified via Ni-affinity chromatography as described above.

### Protein expression and purification for crystallization

For crystallization, Mb and Mb(H64G,V68A) were cloned in HisTag-free form in the Nde *I*/Xho*I* cassette of a pET22 vector. After transformation, C41(DE3) cells were grown in a TB medium (ampicillin, 100 mg L$^{-1}$) at 37 °C (200 rpm) until OD$_{600}$ reached 1.0–1.2. Cells were then induced with 0.25 mM β-d-1-thiogalactopyranoside (IPTG) and 0.3 mM δ-aminolevulinic acid (ALA). After induction, cultures were shaken at 180 rpm and 27 °C and harvested after 18–20 h by centrifugation at 4000 rpm at 4 °C. After cell lysis by sonication, the proteins were purified by strong cationic exchange chromatography, followed by size exclusion chromatography. The lysate was transferred to a column of GE Healthcare SP Sepharose Fast Flow resin equilibrated with 5 mM KPi buffer, pH 7. The resin was washed with 5 column volumes (CV) of 5 mM KPi buffer (pH 7) and then with 5 CV of 10 mM KPi buffer (pH 7). Proteins were eluted with 50 mM KPi buffer (pH 7). After concentration using a 10 kDa Centricon filter, proteins were further purified via gel filtration chromatography using a Superdex 75 10/300 GL column and isocratic elution with 50 mM KPi buffer (pH 7) at a flow rate of 0.5 mL/min. Protein samples for crystallization trials were buffer exchanged with 20 mM Tris buffer (pH 8.4) containing 1 mM EDTA. The imidazole-bound Mb(H64G,V68A) samples were prepared using a similar overall procedure after incubating the protein with KPi buffer containing 10 mM imidazole prior to the two-step purification process.

### UV-Vis measurements

All spectra were recorded anaerobically in 50 mM potassium phosphate buffer at pH 7.0. All the samples with ferrous Mb were prepared and measured anaerobically. Enzyme and reactant concentrations were chosen as indicated in each experimental section.

### Reactions with purified protein (high conc.)

Protein experiments were carried out at a 200 μL scale using 3 mM myoglobin and 30 mM sodium dithionite inside an anaerobic chamber. All Mb solutions were degassed in a sealed vial by purging the headspace with inert gas multiple times via Schlenk line. In a typical procedure, a buffered solution containing myoglobin was charged to another vial inside an anaerobic chamber, and then 10 μL of sodium dithionite solution (200 mM stock solution in 50 mM KPi pH 7) was added. Reactions were initiated by the addition of 10 μL of styrene (a stock solution of 20x desired concentrations in ethanol), followed by the addition of 10 μL of diazoketone (a stock solution of 20x desired concentrations in ethanol). The sealed reaction mixture was left under magnetic stirring for the specified amount of time at room temperature inside an anaerobic chamber. The reactions were prepared for GC analysis (see "Analytical methods" for details) by adding 10 μL of internal standard (50 mM benzodioxole in ethanol) to the reaction mixture, followed by extraction with 400 μL of dichloromethane (DCM). Samples were analyzed by chiral GC as described in the Supplementary Information (see "Analytical methods").

### Reactions with purified protein (low conc.)

Protein experiments were carried out at a 400 μL scale using 20 mM myoglobin and 10 mM sodium dithionite inside an anaerobic chamber. In a typical procedure, a buffered solution containing the myoglobin variant was carefully charged to a vial inside an anaerobic chamber. After diluting the catalyst solution with sodium borate buffer (50 mM, pH 9.0), 20 μL of sodium dithionite solution (200 mM stock solution in 50 mM pH 9 sodium borate buffer) was added. Reactions were initiated by the addition of 20 μL of styrene (from 400 mM stock solution in ethanol), followed by the addition of 20 μL of diazoketone (from 100 mM stock solution in ethanol). The reaction mixture was left under magnetic stirring for 16 h at room temperature inside an anaerobic chamber. The reaction mixtures were added with 20 μL of internal standard (50 mM benzodioxole in ethanol), followed by extraction with 400 μL of dichloromethane. Samples were analyzed by chiral GC as described in the Supplementary Information (see "Analytical methods").

### Protein crystallography

Mb(H64G,V68A) at 2.5 mM (Mb-cIII structure) or 3.6 mM (Mb-imi structure) in 20 mM Tris-HCl pH 8.4, 1 mM EDTA, was mixed with an equal volume of reservoir solution (200 mM Tris-HCl pH 8.9 and 2.4 M (Mb-cIII) or 2.46 M (Mb-imi) ammonium sulfate) to perform hanging-drop vapor diffusion crystallization over a total reservoir volume of 1 mL. To assist in crystal growth, streak seeding from previously grown variant crystals was performed using a Hampton seeding tool (Hampton Research, HR8-133). For Mb-cIII, a crystal was transferred from the drop it was grown in to another 4 μL drop containing well solution with the addition of 10% glycerol. The drop was overlaid with 4 μL mineral oil (MP Biomedicals, 194836). A micropipette tip was dipped in sodium dithionite powder and the tip was then slowly submerged into the drop to allow the dithionite crystals to be released. Then, 1 μL of BDK (2) at 500 mM (dissolved in isopropanol) was added to the drop for a final concentration of 100 mM BDK and incubated for an additional 22.5 min. The crystal was then looped and flash-frozen in liquid nitrogen. Although the crystal was harvested in aerobic conditions in the presence of dithionite, similar structures of Mb-cIII could be obtained with crystals grown in an anaerobic Coy glove bag without the addition of sodium dithionite. For the Mb-imi structure, the crystal was dunked in a cryoprotection solution (reservoir buffer supplemented with 9% sucrose (w/v), 2% glucose (w/v), 8% glycerol (v/v), and 8% ethylene glycol (v/v)) prior to being flash-frozen in liquid nitrogen.

Crystallographic diffraction data was collected at the Stanford Synchrotron Radiation Lightsource beamline 9-2 (Mb-cIII) or the Advanced Photon Source beamline 24-ID-C (Mb-imi). In crystallo spectroscopic data were collected at beamline 9-2 at the Stanford Synchrotron Radiation Lightsource. Crystallographic data were integrated and scaled with XDS[45] and merged using AIMLESS[46]. The PDB 6M8F structure was used for molecular replacement by rigid body refinement using PHENIX for both structures. Restraints file for the Mb-cIII complex was generated using the Grade web server (grade.globalphasing.org). The structures were refined using Coot[47] and PHENIX[48]. For the 8ESS structure, anisotropic refinement was used for protein atoms and isotropic refinement for ligand and ordered solvent atoms. For the 8ESU structure, anisotropic refinement was used for all atoms except for the imidazole and two sulfates that were manually added at a later stage and refined isotropically. All structural figures and maps were made using PyMOL Molecular Graphics System (Schrödinger, LLC) and PHENIX, respectively[48].

## Mössbauer spectroscopy experiments

Mössbauer samples were prepared at a 500 µL scale using 3 mM Mb-imi in 50 mM KPi (pH 7) and 30 mM sodium dithionite inside an anaerobic chamber. Prior to use, the protein samples were degassed in a sealed vial by purging the headspace with inert gas multiple times via Schlenk line. In a typical procedure, a buffered solution containing Mb-imi was charged to Delrin cups inside an anaerobic chamber, and then 20 µL of sodium dithionite solution (750 mM stock solution in 50 mM KPi pH 7) was added. After slow manual stirring for 30 s, 20 µL of BDK (from a 20x stock solution in ethanol) was added and freeze-trapped in liquid nitrogen. Manual or mechanical stirring during the reagent incubation was carried out to ensure homogeneity of the reaction solution. Solution samples were loaded into Delrin cups and subsequently freeze-quenched in liquid nitrogen. Low-temperature $^{57}$Fe Mössbauer spectroscopic analysis was performed using a Janis SVT-400T N2 cryostat for analysis at 80 K. Isomer shift values were calibrated against an α-Fe standard at 298 K. Fitting of the Mössbauer data was performed using WMoss (See Co.) software. The corresponding errors in the fit analyses include the following: $\delta \pm 0.02$ mm/s, $|\Delta E_Q| \pm 3\%$, multicomponent fit quantitation error of $\pm 3\%$. Only zero-field Mössbauer measurements were collected, hence all quadrupole splitting parameters are reported as absolute values. Catalytic solution Mössbauer samples were prepared following the literature protocol for the catalysis and rapid freeze-trapping at specified time points.

## Heme extraction experiments

The samples were prepared in an anaerobic chamber at a 1 mL scale using 400 µM Mb variant in 50 mM potassium phosphate buffer (pH 7) and 4 mM $Na_2S_2O_4$. Then, 20 µL of BDK (from stock solution at 100 mM) was added to achieve a 2 mM BDK concentration. Then, 100 µL aliquots were taken at different time points and extracted with 500 µL of acetone containing 5 mM HCl. The extraction solutions were centrifuged (14,000 rpm, 30 s) and analyzed by MALDI-TOF mass spectrometry using a TiO$_2$ matrix.

## Stopped-flow UV-vis analyses

Kinetic experiments were performed using an Applied Photophysics SX20 stopped-flow UV/vis spectrometer equipped with a double mixing system, direct coupled photodiode array, and anaerobic apparatus. For the reduction experiments, changes in the intensity of the Soret band were monitored upon 1:1 mixing of the Mb variant solution at 10 µM with sodium dithionite solutions at different concentrations (1 to 20 mM in 50 mM KPi pH 7) under anaerobic conditions. To obtain observed rates, the observed decrease in absorbance at 430 nm was fitted to an exponential decay using Pro-KIV software. The rate constant was determined by plotting the fitted exponential rates against the sodium dithionite concentrations (Supplementary Fig. 3).

## Kinetics of Mb-cI formation

The kinetics of the reaction of Mb with BDK (**2**) to form the **Mb-cI** complex were analyzed via stopped-flow UV-vis spectroscopy. Changes in intensity of the Soret band were monitored upon 1:1 mixing of anaerobic Mb solutions at 10 µM with anaerobic BDK solutions at different concentrations (0.06 to 1 mM in 20% EtOH 50 mM KPi pH 7). All solutions of Mb had been pre-treated with 10 mM $Na_2S_2O_4$ no more than 10 min before the experiment. To obtain observed rates, the observed decrease in absorbance at 425 nm was fitted to an exponential decay using Pro-KIV software. The rate constant was determined by plotting the fitted exponential rates against the BDK concentrations (Fig. 4).

## Kinetics of Mb-cIII formation

The kinetics of the reaction of Mb with BDK (2) to form Mb-cIII complex were analyzed via stopped-flow UV-vis spectroscopy. Changes in intensity of the Soret band were monitored upon 1:1 mixing of anaerobic Mb solutions at 10 µM with anaerobic BDK solutions of different concentrations (1 to 10 mM in 20% EtOH 50 mM KPi pH 7). All solutions of Mb had been pre-treated with 10 mM $Na_2S_2O_4$ no more than 10 min before the experiment. To extract rates, the observed decrease in absorbance at 425 nm was fitted to an exponential decay using Pro-KIV software. The rate constant was determined by plotting the fitted exponential rates against the BDK concentrations (Fig. 4).

## Kinetic analysis of Mb-cIII formation in the presence of styrene

The kinetics of the reaction of Mb with BDK (**2**) in the presence of styrene were analyzed via stopped-flow UV-vis spectroscopy. Changes in intensity of the Soret band were monitored upon 1:1 mixing of anaerobic Mb solutions at 10 µM containing styrene at different concentrations (3 to 20 mM 10% EtOH in 50 mM KPi pH 7) and anaerobic 1 mM BDK solution containing 10% EtOH in 50 mM KPi pH 7. All solutions of Mb had been pre-treated with 10 mM $Na_2S_2O_4$ no more than 10 min before the experiment. Full spectra at 3 mM and 20 mM are shown in Fig. 4c, d.

## Kinetics of Mb-imi regeneration from Mb-cIII

The degradation of Mb-cIII to generate Mb-imi was analyzed by UV-Vis spectrometer. The reaction was carried out in an anaerobic chamber at a high concentration of protein (3 mM) and a 10 mL aliquot of the reaction mixture was diluted to 1 mL with 50 mM KPi pH 7 to be monitored by UV-Vis spectrometer (Supplementary Fig. 5).

## Reporting summary

Further information on research design is available in the Nature Portfolio Reporting Summary linked to this article.

## Data availability

Data supporting the findings of this study are included in this published article and the supplementary information files. Additional datasets generated during and/or analyzed during the current study are available from the corresponding author upon reasonable request. Protein crystal structures reported in this manuscript have been deposited in the Protein Data Bank (PDB) under accession codes 8ESS and 8ESU.

## Code availability

No custom computer codes or mathematical algorithms have been used for this research.

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

## Acknowledgements

This work was supported by the U.S. National Institute of Health grants GM098628 (R.F.) and GM111480 (M.L.N.), U.S. National Science Foundation grants CBET-1929256 (R.F.) and CHE-2054897 (Y.Z.), and startup funds from Cornell University (N.A.). R.F. acknowledges support from the Cancer Prevention and Research Institute of Texas (CPRIT RR230018) and Robert A. Welch Foundation (Chair, AT-0051). MS instrumentation at the University of Rochester is supported by U.S. National Institute of Health grant S10OD030302. Use of the Stanford Synchrotron Radiation Lightsource, SLAC National Accelerator Laboratory, is supported by the U.S. Department of Energy, Office of Science, Office of Basic Energy Sciences under Contract No. DE-AC02-76SF00515. The SSRL Structural Molecular Biology Program is supported by the DOE Office of Biological and Environmental Research, and by the National Institutes of Health, National Institute of General Medical Sciences (P30GM133894). The contents of this publication are solely the responsibility of the authors and do not necessarily represent the official views of NIGMS or NIH. This work is based upon research conducted at the Northeastern Collaborative Access Team beamlines, which are funded by the National Institute of General Medical Sciences from the National Institutes of Health (P30 GM124165). This research used resources from the Advanced Photon Source, a U.S. Department of Energy (DOE) Office of Science User Facility operated for the DOE Office of Science by Argonne National Laboratory under Contract No. DE-AC02-06CH11357.

## Author contributions

D.N., R.F., J.-P.B. and N.A. conceived the project and designed the experiments; D.N. performed and analyzed the stopped-flow UV-vis experiments, Mössbauer experiments, and enzymatic reactions under the supervision of R.F.; J.-P.B. performed the crystallization experiments and solved the crystal structures under the supervision of N.A.; R.L.K. and Y.W. performed the computational analysis under the supervision of Y.Z.; M.C.A. performed the Mössbauer experiments under the supervision of M.L.N.; J.D.V. contributed to the kinetic experiments; D.N., J.-P.B., Y.Z. and R.F. wrote the manuscript. All authors discussed the results and contributed to the final manuscript.

## Competing interests

The authors declare no competing interests.
