## [Peer Review File · Nature Communications]

Mechanistic Manifold in a Hemoprotein-Catalyzed Cyclopropanation Reaction with DiazoketoneEditorial Note: This manuscript has been previously reviewed at another journal that is not operating a transparent peer review scheme. This document only contains reviewer comments and rebuttal letters for versions considered at Nature Communications .

REVIEWERS' COMMENTS

Reviewer #1 (Remarks to the Author):

The revision by Fasan, collaborators, and coworkers seems suitable for Nature Communications. My concerns with the original submission pertained to the level of novelty, and Nat. Commun. Is a reasonable alternative medium for publication. Other reviewers can comment on the degree to which the author has made sufficient revisions to address their comments on the computations.

My only comment is a minor one and easy to address, although it would be reasonable address with additional experimentation. The authors state that "Mb-imi, exhibits identical reactivity in the cyclopropanation reaction as the free enzyme" in the presence or absence of imidazole in varying concentrations. While Table S1 shows identical full conversions, this is a mechanistic paper and an effect of the imidazole on the rate is really what I was probing. Table 1 does not provide such rate data. "Identical reactivity" would mean identical rates, and the Im should inhibit the reaction if there is reversible exchange of Im with the carbene, but a lack of inhibition if this exchange is irreversible. The authors did not address this question in the response and did not reword this sentence to reflect what the data actually show.

My other comments were addressed or were more general ones about whether the novelty warranted publication in the journal to which the manuscript was originally submitted.

Reviewer #4 (Remarks to the Author):

Most of my previous concerns have been addressed or answered in the response letter. In my opinion, the paper in its revised form is suitable for publication.

Reviewer #5 (Remarks to the Author):

My questions were fully addressed and concerns resolved in a revised manuscript. The results are suitable for Nature Communications. I recommend accepting manuscript for publication.

Reviewer #6 (Remarks to the Author):

The response to Reviewer 3 is hard to evaluate because I don't have the original manuscript thus I cannot gauge the changes made. The manuscript in its present form is clear and the claims made by the authors are well supported. The Reviewer 3 comment regarding the ICP (Iron-porphyrin carbene complex) was addressed by taking away the Mössbauer spectrum of the ICP intermediate in figure SI-10.

I would encourage the authors to take a different action. I would suggest that the authors move their commentary on the ICP intermediate in the Results section and include the Mössbauer spectrum in the SI. The authors say that the ICP is not essential to the manuscript. Nevertheless, because the authors present DFT calculations of the Mössbauer parameters for this intermediate, the Mössbauer experimental data are also important to be reported. In addition, in my opinion this Fe-porphyrin carbene is extremely interesting in its own right for the larger chemistry community.

Reviewer #1:

Comments: *The revision by Fasan, collaborators, and coworkers seems suitable for Nature Communications. My concerns with the original submission pertained to the level of novelty, and Nat. Commun. Is a reasonable alternative medium for publication. Other reviewers can comment on the degree to which the author has made sufficient revisions to address their comments on the computations.*

My only comment is a minor one and easy to address, although it would be reasonable to address with additional experimentation. The authors state that “Mb-imi, exhibits identical reactivity in the cyclopropanation reaction as the free enzyme” in the presence or absence of imidazole in varying concentrations. While Table S1 shows identical full conversions, this is a mechanistic paper and the effect of the imidazole on the rate is really what I was probing. Table 1 does not provide such rate data. “Identical reactivity” would mean identical rates, and the I_m should inhibit the reaction if there is reversible exchange of I_m with the carbene, but a lack of inhibition if this exchange is irreversible. The authors did not address this question in the response and did not reword this sentence to reflect what the data show.

My other comments were addressed or were more general ones about whether the novelty warranted publication in the journal to which the manuscript was originally submitted.

Response and changes: We thank Reviewer #1 for their comments. We agree about the importance of assessing the impact of imidazole on the enzyme kinetics. In response to the reviewer's request, we measured the initial rate of cyclopropanation with benzodiazoketone (BDK) and styrene in the presence of different concentrations of imidazole. As shown by new data included in the Supplementary Figure 6, no significant change in initial rates of cyclopropanation was observed at different imidazole concentrations, supporting the lack of inhibition and conclusions made in the manuscript.

Reviewer #6:

***Comment:** The response to Reviewer 3 is hard to evaluate because I don't have the original manuscript thus I cannot gauge the changes made. The manuscript in its present form is clear and the claims made by the authors are well supported. The Reviewer 3 comment regarding the ICP (Iron-porphyrin carbene complex) was addressed by taking away the Mössbauer spectrum of the ICP intermediate in figure SI-10.*

I would encourage the authors to take a different action. I would suggest that the authors move their commentary on the ICP intermediate in the Results section and include the Mössbauer spectrum in the SI. The authors say that the ICP is not essential to the manuscript. Nevertheless, because the authors present DFT calculations of the Mössbauer parameters for this intermediate, the Mössbauer experimental data are also important to be reported. In addition, in my opinion this Fe-porphyrin carbene is extremely interesting in its own right for the larger chemistry community.

Answer: The species in question was another intermediate other than the IPC complex. We tentatively attributed this species to the initial diazo-heme complex. Although the Mossbauer spectrum of the trapped species was of excellent quality and it showed a species distinct from those currently presented in the paper, trapping of that species turned out to be very difficult and could not be reproduced, as noted in the original manuscript. Following reviewer's #3 request, description and discussion of those results were removed from the manuscript.